# Symbiont Identity Impacts the Microbiome and Volatilome of a Model Cnidarian-Dinoflagellate Symbiosis

**DOI:** 10.3390/biology12071014

**Published:** 2023-07-17

**Authors:** Maggie Wuerz, Caitlin A. Lawson, Clinton A. Oakley, Malcolm Possell, Shaun P. Wilkinson, Arthur R. Grossman, Virginia M. Weis, David J. Suggett, Simon K. Davy

**Affiliations:** 1School of Biological Sciences, Victoria University of Wellington, Wellington 6012, New Zealand; 2Climate Change Cluster, University of Technology Sydney, Sydney Broadway, Sydney, NSW 2007, Australia; 3School of Environmental and Life Sciences, University of Newcastle, Callaghan, NSW 2308, Australia; 4School of Life and Environmental Sciences, University of Sydney, Sydney, NSW 2006, Australia; 5Wilderlab Ltd., Wellington 6022, New Zealand; 6Carnegie Institution for Science, Department of Plant Biology, Stanford, CA 94305, USA; 7Department of Integrative Biology, Oregon State University, Corvallis, OR 97331, USA; 8KAUST Reefscape Restoration Initiative (KRRI) and Red Sea Research Center (RSRC), King Abdullah University of Science and Technology, Thuwal 23955, Saudi Arabia

**Keywords:** Aiptasia, Symbiodiniaceae, volatilomics, BVOC, coral

## Abstract

**Simple Summary:**

The success of coral reefs is underpinned by the symbiosis between corals and their dinoflagellate symbionts. Crucially, metabolic interactions between the two partners support coral metabolism and survival, although these can be influenced by symbiont identity and inter-partner compatibility. Here, we measured how symbiont identity influences the release of biogenic volatile organic compounds (BVOCs) via the symbiosis, and related this to concurrent shifts in the host microbiome; BVOCs are end-products of metabolism and important biological signal molecules. We used the sea anemone Aiptasia, a model system for cnidarian-dinoflagellate symbiosis, when either symbiont-free, populated with its native symbiont, or populated with a non-native symbiont. We detected 142 BVOCs across all treatments. The volatile profiles of symbiont-free anemones and those containing the native symbiont were distinct, while the volatile profile of anemones containing the non-native symbiont shared characteristics with both. The symbiotic state also caused a change in the host microbiome, but this did not explain the changes seen in BVOC release. These findings contribute to our understanding of how corals may respond to climate change should they acquire novel symbionts post-bleaching. Furthermore, we provide a platform for future studies of the metabolic and/or signalling roles of BVOCs in this important symbiosis.

**Abstract:**

The symbiosis between cnidarians and dinoflagellates underpins the success of reef-building corals in otherwise nutrient-poor habitats. Alterations to symbiotic state can perturb metabolic homeostasis and thus alter the release of biogenic volatile organic compounds (BVOCs). While BVOCs can play important roles in metabolic regulation and signalling, how the symbiotic state affects BVOC output remains unexplored. We therefore characterised the suite of BVOCs that comprise the volatilome of the sea anemone *Exaiptasia diaphana* (‘Aiptasia’) when aposymbiotic and in symbiosis with either its native dinoflagellate symbiont *Breviolum minutum* or the non-native symbiont *Durusdinium trenchii*. In parallel, the bacterial community structure in these different symbiotic states was fully characterised to resolve the holobiont microbiome. Based on rRNA analyses, 147 unique amplicon sequence variants (ASVs) were observed across symbiotic states. Furthermore, the microbiomes were distinct across the different symbiotic states: bacteria in the family Vibrionaceae were the most abundant in aposymbiotic anemones; those in the family Crocinitomicaceae were the most abundant in anemones symbiotic with *D. trenchii*; and anemones symbiotic with *B. minutum* had the highest proportion of low-abundance ASVs. Across these different holobionts, 142 BVOCs were detected and classified into 17 groups based on their chemical structure, with BVOCs containing multiple functional groups being the most abundant. Isoprene was detected in higher abundance when anemones hosted their native symbiont, and dimethyl sulphide was detected in higher abundance in the volatilome of both Aiptasia-Symbiodiniaceae combinations relative to aposymbiotic anemones. The volatilomes of aposymbiotic anemones and anemones symbiotic with *B. minutum* were distinct, while the volatilome of anemones symbiotic with *D. trenchii* overlapped both of the others. Collectively, our results are consistent with previous reports that *D. trenchii* produces a metabolically sub-optimal symbiosis with Aiptasia, and add to our understanding of how symbiotic cnidarians, including corals, may respond to climate change should they acquire novel dinoflagellate partners.

## 1. Introduction

Coral reefs are highly diverse marine ecosystems and provide habitats and food for a wealth of marine life [1]. At the core of these diverse ecosystems is the symbiotic relationship between dinoflagellate algae (family: Symbiodiniaceae) and their cnidarian hosts [2,3]. Dinoflagellate symbionts are housed intracellularly within cnidarian gastrodermal cells, where photosynthetic products are translocated from the symbiont to the cnidarian host, thereby sustaining host metabolism, growth, reproduction, and ultimately survival in a resource-poor environment [4]. In return, the dinoflagellate is supplied with inorganic nutrients such as nitrogen and phosphorus [2], and a stable habitat in which to photosynthesise in the absence of predators. The nature of this symbiosis has been intensively researched for decades, not only because of its importance for maintaining the foundation of reef ecosystems, but also because of the vulnerability of the symbiosis to anthropogenic stressors [5,6], particularly increased ocean temperatures as a consequence of climate change [7,8].

Comprised of taxonomically distinct genera and species [9,10,11], the Symbiodiniaceae family is also physiologically diverse, in particular with respect to photo- and nutritional physiology, cellular growth, and stress tolerance [12,13]. While many cnidarian hosts exhibit specificity for the species of Symbiodiniaceae with which they are naturally associated [14], others can be associated with a variety of symbiont species [15]. The species of Symbiodiniaceae with which a cnidarian naturally associates is called the native, or homologous, symbiont. Following bleaching, hosts can acquire novel symbiont taxa from the environment [16], or more commonly be repopulated by symbiont taxa that were previously less abundant within their tissues [17]; these changes in the symbiont population are termed “switching” and “shuffling”, respectively. While repopulation or reorganisation with thermally tolerant symbionts can help corals withstand changing environmental conditions, it can also result in an association with altered, suboptimal nutritional exchange and differences in the activities of specific metabolic pathways [18,19]. Such changes in the metabolic characteristics of cnidarians associated with non-native (heterologous) symbionts, relative to associations with homologous symbionts, have been documented through differences in the transcriptome [18], proteome [20,21], and metabolome [18,22].

In addition to algal symbionts, cnidarians are also associated with an array of other microorganisms, including bacteria, fungi, viruses, and archaea [23,24]. The association with these microorganisms is crucial for the optimal functioning of coral reefs, and—collectively with algal associates—contributes to the physiological functions of the metabolically networked metaorganism, referred to as a ‘holobiont’. This aggregation of prokaryotes and eukaryotes has a dynamic influence on holobiont function, and community composition will reflect the current environment [25]. The importance of bacteria in the overall fitness of the coral holobiont led to the proposal of the ‘coral probiotic hypothesis’ [26], postulating that the microbiome composition is optimised to support coral biology given current environmental conditions. Indeed, it has been shown that cnidarian-associated bacteria play an important role in holobiont health, contributing to pathogen defence, metabolism, and nutrient cycling [27,28,29,30], and that host microbiota differ in symbiotic versus aposymbiotic anemones [31]. Due to their rapid generation time, bacterial populations can quickly shift, selecting communities beneficial to the host [30,32]. For example, microbiome flexibility is thought to play a role in the tolerance of the coral *Fungia granulosa* to high salinity levels [33] and the coral *Acropora hyacinthus* to thermal stress [34]. Microbiome composition is thought to be so important to the success of the coral holobiont community that it has been proposed that microbiome transplants between cnidarians may aid a holobiont’s ability to adjust to changing environmental conditions [35].

Biogenic volatile organic compounds (BVOCs) are low-molecular-weight (<200 Da) chemicals with high vapour pressure [36] that are produced by a broad diversity of organisms. BVOCs are emitted by all organisms, including bacteria [37], fungi [38], algae [39], plants [40], insects [41], corals [42], sea anemones [43], and mammals [44]. On a global scale, the emission of organically produced VOCs exceeds that of VOCs from anthropogenic sources [45]. BVOCs can be metabolic by-products [46], end-products (e.g., serving as pollination attractants [47]), or precursors to other molecules (e.g., plant hormones) [48]. While the majority of BVOC research has focused on compounds produced by terrestrial ecosystems [49], aquatic ecosystems also produce a diversity of BVOCs that are collectively designated as ‘volatilomes’ [50]. The composition and abundance of BVOC production has been shown to vary among environmental conditions and across different species of Symbiodiniaceae [39] and corals [42], but the physiological roles of many of these compounds remain unknown. We hypothesise that, as with previously described classes of organic compounds, BVOCs will show altered abundance in symbiotic associations with heterologous partners. 

The symbiotic sea anemone *Exaiptasia diaphana* (‘Aiptasia’) is a model system for studying cnidarian-dinoflagellate symbiosis [51,52]. Like corals, Aiptasia forms a stable symbiosis with dinoflagellates in the family Symbiodiniaceae [53]. Aiptasia can be rendered aposymbiotic via laboratory-induced bleaching, maintained in this state for years through heterotrophic feeding and repopulated by a variety of symbiont species [54,55]. Furthermore, Aiptasia can reproduce asexually, allowing the maintenance of clonal populations [56]. While previous work has shown how the microbiome and volatilome change in Aiptasia in response to symbiosis with a homologous symbiont [31,43] and in response to thermal stress [57], there is a gap regarding the impact of symbiont identity on the microbiome and BVOC production. We therefore explored how the microbiome and volatilome change in response to a switch in the dinoflagellate endosymbiont community of Aiptasia. 

We specifically profiled Aiptasia-associated BVOCs in three different symbiotic states: (1) in symbiosis with the native partner *Breviolum minutum* (homologous symbiont); (2) in symbiosis with a non-native but thermally tolerant partner, *Durusdinium trenchii* (heterologous symbiont); and (3) in the absence of symbionts (aposymbiosis). The microbial communities present in the holobiont of these three states were also assessed, to further elucidate how altering the predominant algal symbiont influences holobiont composition and the net outcome of BVOC emissions. In doing so, we provide a further understanding of how holobionts may respond to changing environmental conditions and establish a platform for the future elucidation of the roles of BVOCs in symbiotic cnidarians.

## 2. Methods

### 2.1. Experimental Organisms

A long-term (15+ years) clonal culture of the sea anemone Aiptasia (culture ID: NZ1) of unknown Pacific origin [18] was maintained in the laboratory in 0.22 µm filtered seawater (FSW) at 25 °C and approximately 70 µmol photons m^−2^ s^−1^ on a 12:12 h light dark cycle (GE Lighting T5 F54W/840). Clonal anemones (n = 100) were rendered aposymbiotic (i.e., symbiont–free) using menthol-induced bleaching: exposure to menthol (20% *w*/*v* in ethanol; Sigma-Aldrich, Auckland, New Zealand) at a final concentration of 0.19 mmol L^−1^ in 0.22 µm FSW [54]. Anemones were incubated in menthol for 8 h during the 12 h light period, and after which, photosynthesis was inhibited by replacing menthol/FSW with FSW containing 5 µmol L^−1^ 3-(3,4–dichlorophenyl)-1,1-dimethylurea (DCMU; 100 mmol L^−1^ dissolved in EtOH, Sigma-Aldrich) for 16 h to prevent repopulation by inhibiting the photosynthesis of the remaining symbionts. After repeating this 24 h cycle for four consecutive days, anemones were maintained in 0.22 µm FSW for three days. Anemones were fed once weekly with *Artemia* sp. nauplii, with fresh FSW changes 8 h post-feeding. This protocol was continued for six weeks, and aposymbiotic anemones were maintained in the dark at 25 °C for 1.5 years prior to BVOC sampling. 

*Breviolum minutum* (ITS2 type B1, culture ID ‘FLAp2’) was used as the homologous symbiont and *Durusdinium trenchii* (ITS2 D1a, culture ID ‘Ap2’) was the heterologous symbiont. *B. minutum* and *D. trenchii* isolates were grown in 0.22 µm FSW enriched with f/2-medium maintained at 25 °C in a climate-controlled incubator. Symbiodiniaceae cultures were grown under light provided by fluorescent lamps (Osram Dulux 36/W890) at approximately 70 µmol photons m^−2^ s^−1^ on a 12:12 h light/dark cycle. One week prior to inoculation, cultures were diluted with fresh f/2 medium to ensure that they were in exponential growth. 

Symbiotic anemones were generated by recombining aposymbiotic anemones with cultured Symbiodiniaceae isolates. Prior to this inoculation, aposymbiosis was confirmed using fluorescence microscopy (Olympus IX53 inverted microscope; 100× magnification). A subset of aposymbiotic anemones were starved for seven days prior to inoculation with either cultured *B. minutum* or *D. trenchii* (n = 25 for both species). An aliquot (~20 µL) of Symbiodiniaceae culture, concentrated via centrifugation to a density of 3 × 10^6^ cells mL^−1^, was pipetted directly onto the oral disc of individual aposymbiotic anemones. *Artemia* sp. nauplii were mixed into this suspension to encourage the phagocytosis of algal cells [58]. Inoculated anemones were fed twice weekly with *Artemia* sp. nauplii and maintained at 25 °C and approximately 70 µmol photons m^−2^ s^−1^ (GE Lighting T5 F54W/840) on a 12:12 h light/dark cycle. Anemones were fully symbiotic for six months prior to BVOC sampling. The presence of intracellular symbionts was confirmed via fluorescence microscopy three months prior to sampling, as described above. Symbiosis was maintained and inspected visually weekly until sampling, and after which, the symbiont cell densities were determined as described below. 

Symbiotic anemones were maintained at a constant temperature of 25 °C and light intensity of 70 µmol photons m^−2^ s^−2^ on a 12:12 h light/dark cycle as described above. Aposymbiotic anemones were maintained at a constant temperature of 25 °C and kept in the dark. All anemones were fed twice weekly, with water changes the day after feeding, using 0.22 µm FSW. All animals were retained in these states for six months prior to BVOC sampling. On the day of sampling, the maximum quantum yield of photosystem II (*F_v_/F_m_*, dimensionless) was used as a relative indicator of photosynthetic competency in symbiotic anemones (Appendix A). Symbiotic anemones were dark-acclimated for 15 min before performing measurements using an Imaging Pulse Amplitude Modulated Fluorometer (I-PAM, Walz, Effeltrich, Germany; settings: measuring light = 4, saturation intensity = 8, saturation width = 0.8 s, gain = 3, and damping = 3).

### 2.2. Microbe Sampling and Microbiome Analysis

Frozen Aiptasia samples from which BVOCs were analysed (see below; ~10–15 animals per sample, depending on symbiotic state) were thawed and resuspended in sterile artificial seawater (ASW) to produce 1 mL samples, and pooled anemones were mechanically homogenised for 30 s on ice. DNA extraction was performed, and subsequent 16S rRNA diversity was analysed from 15 samples across three symbiotic states: aposymbiotic anemones (n = 5; ~15 anemones/vial), anemones symbiotic with their homologous symbiont, *B. minutum* (n = 5; ~10 anemones/vial), and anemones symbiotic with a heterologous symbiont, *D. trenchii* (n = 5; ~12 anemones/vial). 

DNA extraction from 300 µL of anemone homogenate was performed using the DNeasy Plant Mini Kit (Quiagen), according to the manufacturer’s instructions. DNA concentrations were quantified using a NanoDrop spectrophotometer (Impen NanoPhotometer^TM^ NP80, Thermo Fisher). Variable regions 5 and 6 of the 16s rRNA gene were targeted using the primer pair 784F [5′ TCGTCGGCAGCGTCAGATGTGTATAAGAGACAG–AGGATTAGATACCCTGGTA 3′] and 1061 R [5′ GTCTCGTGGGCTCGGAGATGTGTATAAGAGACAG–CRRCACGAGCTGACGAC 3′] with Illumina adaptor overhangs (underlined above). This primer pair was used in previous cnidarian microbiome studies [30,33]. Extracted DNA was sent to the University of Auckland Genomics Facility for both amplification of the variable rRNA regions and sequencing of the amplification products (Auckland, New Zealand). gDNA was normalised to 50 ng µL^−1^ and 200 ng were used as an input for the 25 µL PCR reaction. The PCR mix included 12.5 µL of 2× Platinum SuperFi Master Mix. The concentration of each primer in the reaction was 0.2 µM. The first round of PCR thermal cycling conditions was 3 min initial denaturation at 95 °C followed by 25 cycles of 30 s at 95 °C, 30 s at 55 °C, and 30 s at 72 °C, followed by a final elongation step at 72 °C for 5 min. After the first round of PCR, 1 µL of each sample was checked for quality using a BioAnalyzer (2100-Agilent, Santa Clara, CA, USA) with the HS DNA chip. A 0.8× volume of AMPure XP beads was used to clean up the reactions (20 µL PCR product, 16 µL bead suspension) with two ethanol washes and elution in 12 µL molecular grade water. A total of 1.5 µL of the cleaned first-round PCR product was used as an input to the second-round indexing PCR, which used Nextera V2 indexing primers and the Platinum SuperFi MasterMix. Thermal cycling conditions were 3 min at 95 °C, followed by 8 cycles of 30 s at 95 °C, 30 s at 55 °C, and 30 s at 72 °C, followed by 5 min at 72 °C. After this round of amplification, dsDNA was quantified using a Qubit dsDNA HS (high-sensitivity) Assay Kit. A total of 10 µL of each sample were pooled and cleaned up with two rounds of AMPure beads using a 1:1 ratio of beads to sample. Samples were gently mixed and incubated at room temperature for 5 min, and after which, the supernatant was discarded. The same beads were resuspended with fresh 80% ethanol twice. After both ethanol cleanup steps, the supernatant was discarded and the beads were allowed to air dry for 10 min. Once dry, the beads were resuspended with 52.5 µL of 10 mM Tris pH 8.5 and incubated at room temperature for two minutes. Samples were then loaded onto the Illumina MiSeq platform for sequencing, according to the manufacturer’s protocol [59].

### 2.3. BVOC Sampling and Volatilome Characterisation

Biogenic volatile organic compounds (BVOCs) were collected and analysed from anemones in three different symbiotic states: aposymbiotic anemones (n = 8; ~15 anemones per vial); symbiotic anemones with *B. minutum* (n = 8; ~10 anemones per vial); and symbiotic anemones with *D. trenchii* (n = 8; ~12 anemones per vial). Different numbers of anemones were used across treatments to account for variability in anemone size. The experimental setup and BVOC retrieval were performed using previously established methods described by Wuerz et al. [44] and modified from Lawson et al. [39,42]. The night before BVOC sampling, experimental organisms were transferred into sterile 150 mL glass vials (Wheaton, Millville, NJ, USA) containing 75 mL 0.22 µm FSW, where they were retained under conditions identical to those used for growth. Immediately prior to sampling, FSW was refreshed, and vials were sealed using 20 mm PTFE/Si crimp caps (Agilent, Santa Clara, CA, USA). BVOCs were collected by passing instrument grade air (100 mL min^−1^; BOC Gases, Wellington, New Zealand) into gas-tight sampling vials for 20 min, whereby the outgoing air was passed through open-ended thermal desorption tubes (TDTs; Markes International Ltd., Llantrisant, UK) containing the sorbent Tenax TA, and onto which the BVOCs were adhered. After 20 min, TDTs were immediately sealed using brass storage caps and stored at 4 °C until processing. After BVOC retrieval, anemones were immediately frozen at −80 °C and stored at this temperature until DNA extraction, symbiont cell density determination, and protein quantification. All TDTs were analysed within two weeks of sampling to minimise sample degradation using gas chromatography coupled with single quadrupole mass spectrometry (GC-MS), as per Olander et al. [60]. TDT desorption was performed using a Marks Unity 2 Series Thermal Desorber and ULTRA2 multitube autosampler. Desorption occurred at 300 °C for 6 min, after which, BVOCs were concentrated on a Tenax cold trap at −30 °C. The cold trap was subsequently flash-heated to 300 °C, and the concentrated sample was injected onto a 7890A GC (Agilent Technologies, Ltd., Melbourne, Australia) via a transfer line maintained at 150 °C. The GC was equipped with a 60 m × 0.32 mm BP1 column (Agilent Technologies, Ltd., Melbourne, Australia) with a film thickness of 1 µm. To encourage the complete desorption of BVOCs, the GC oven was heated at 35 °C for 5 min, followed by increasing the temperature to 160 °C at a rate of 4 °C min^−1^, and then raising the temperature to 300 °C at 20 °C min^−1^ and holding it for 5 min. All samples were run spitless at a flowrate of 2.3 mL min^−1^. The GC was attached to a Model 5975 mass-selective detector (Agilent), with the scanning range set to 35–300 amu. 

Compounds from GC-MS were run through the open-source MS data processing program OpenChrom [61] to remove common contaminating ions (amu: 73, 84, 147, 149, 207, and 221). Output files were imported to Galaxy [62] and processed using the metaMS.runGC package (Galaxy version 2.1.1; [63,64,65]) in Workflow4Metabolomics. Peaks were tentatively identified (hereafter ‘identified’) against a database in the National Institute of Standards and Technology (NIST) Mass Spectral library (NIST 14 library in NIST MS Search v2.2; NIST, Gaithersburg, MD). A compound match factor of at least 60% was required for compound identity to be recorded; otherwise, the compound was listed as ‘unknown’. 

Prior to desorption, each TDT was injected with 0.2 µL of 150 ppm bromobenzene (GC grade, Sigma Aldrich, Castle Hill, NSW, Australia) in methanol (HPLC grade, Sigma Aldrich) and, as an internal standard, BVOCs were also collected from filtered (0.22 µm) seawater blanks (n = 8) using the same methods at the time of sample collection. Average values for the BVOCs present in the blanks were subtracted from all samples. Finally, peak abundances were normalised to the protein content of each replicate. Protein was released from the samples via lysis using ultrasonication (VCX500; Sonics & Materials Inc., Newtown, CT, USA) and the protein content was measured using the fluorometric Qubit Protein Assay Kit (Thermo Fisher Scientific, Waltham, MA, USA; [66]). Compounds identified as likely methodological artefacts, e.g., silicon-containing compounds, were also removed from the dataset, as they were suspected contaminants from dimethylpolysiloxane hydrolysis [67]. Only BVOCs present in a minimum of four biological replicates in at least one symbiotic state were classed as ‘present’ in a state. All BVOCs were grouped according to their chemical class. 

### 2.4. Symbiont Cell Density and Protein Determination

Anemones (n = 8, with 10–15 pooled anemones per replicate), frozen as described above, were thawed on ice and homogenised using a saw-tooth homogeniser (Ystral D-79282, Ballrechten-Dottingen, Germany) for 1 min at a mid-speed setting in 500 µL double distilled water (EASYpure II RF/UV ultrapure water system). Host and symbiont fractions were separated via centrifugation at 400× *g* for 2 min, after which, the symbiont fraction was washed of residual host material by one additional resuspension and centrifugation with double distilled water. The supernatant fraction was analysed for protein content using the fluorometric Qubit Protein Assay Kit. Cell counts were performed using a haemocytometer (Improved Neubauer) with eight replicate counts per sample and normalised to protein content. Cell density (symbionts per mg host protein) was then calculated (Appendix A).

### 2.5. Data Analysis

For microbial analysis, the MiSeq output fastq file was de-multiplexed in R [68] using the insect package (v 1.4.0; [69]) and trimmed sequences were filtered to produce a table of exact amplicon sequence variants (ASVs) using the DADA2 R package [70]. ASVs were identified to the lowest possible taxonomic rank using a two-step classification process. This involved the following: (1) exact matching against the RDP v18 reference database (accessed on 15 June 2022 from https://doi.org/10.5281/zenodo.4310150) and assigning taxonomy at the lowest common ancestor level (LCA; i.e., assigning to family level if there were sequence matches with 100% identity to more than one genus), and (2) querying any unwanted sequences against the same RDP reference database using the SINTAX classification algorithm [71] with a conservative assignment threshold of >0.99. In both cases, the maximum assignment resolution was set to genus level due to the high over-classification rates associated with species-level assignment [71]. Sequencing data, along with taxonomic assignment, are publicly available at https://github.com/maggiewuerz/symbiont_identity_microbiome.git (accessed on 23 June 2023). 

Differential abundance of BVOCs was estimated using the *limma* R package [72]; the *voom* function was used to convert counts to log_2_-counts-per-million and to assign weights to each observation based on the mean variance trend. Functions *lmFit*, *eBayes*, and *topTable* were used to fit weighted linear regression models and calculate empirical Bayes moderated t-statistic and q-values [73]. Box plots, bar graphs, and pie charts were created using *ggplot2* [74] in RStudio. Pairs of biological replicates, standardised using the *decostand* function in the *vegan* package [75] in R (version 1.2.5033), were compared using the Bray–Curtis similarity measure, and this output was subjected to non-metric multidimensional scaling to visualise differences among groups based on BVOCs. To compare the dispersion of biological replicates using a distance measure, PERMANOVA was performed to test whether the distance between biological replicates was greater between treatments than within treatments. PERMANOVA was performed on NMDS scores using the *adonis* function, and post hoc tests were performed using the *pairwise.adonis* function in the vegan package in R (version 1.2.5033). PERMANOVA was performed for both volatilomes and microbiomes. To determine whether there was any correlation between microbiomes and volatilomes, Procrustes analysis was performed to compare the distances between these two datasets, using the *procrustes* function on NMDS scores and using the vegan package in R (version 1.2.5033). The differential abundance of microbial taxa was measured using the relative abundance of each microbial taxon within a sample and estimated using the *limma* R package, as described above. 

## 3. Results

### 3.1. Bacterial Community of Aiptasia in Different Symbiotic States

Bacterial taxa identified by 16S diversity data were grouped at the family level, revealing notable differences across the three symbiotic states (Figure 1). Aposymbiotic anemones were generally dominated by Vibrionaceae (average 22% of the microbiome) and Campylobacteraceae (16%), although variability among the samples was high. In contrast, anemones symbiotic with *D. trenchii* were dominated by Crocinitomicaceae (24%) and the class Gammaproteobacteria (mean 17%), and anemones symbiotic with *B. minutum* were dominated by Gammaproteobacteria (20%) and ‘other’ bacteria (unclassified or rare taxa; 28%). Interestingly, a single ASV of Campylobacteraceae was identified exclusively in aposymbiotic anemones. 

Ten microbial ASVs were identified in all samples across the dataset, designated as the ‘core microbiome’ (i.e., ASVs present in all biological replicates). These taxa included the following: *Alteromonas* sp.; Crocinitomicaceae; three taxa within the Gammaproteobacteria; *Maricaulis maris*; *Owenweeksia* sp.; *Pseudoalteromonas arabiensis*; *Rhizobium subbaraonis*; and Rhodobacteraceae (Appendix A).

### 3.2. Contrasting Microbiota between Symbiotic States

A total of 57 bacterial taxa were differentially abundant between the three symbiotic states (Appendix A).

(i)Aposymbiotic anemones versus anemones harbouring homologous symbionts

A total of 41 differentially abundant bacterial taxa were recorded between aposymbiotic anemones and anemones symbiotic with *B. minutum*. Of these, 16 were more abundant in anemones symbiotic with *B. minutum*, including 1 taxon of the genus *Labrenzia*, 2 taxa of the genus *Chlamydiia*, and 2 taxa of the family Alteromonadaceae. A total of 25 ASVs were more abundant in aposymbiotic anemones, including 1 taxon of the family Campylobacteraceae, 8 taxa of the family Vibrionaceae, and 2 species of the family Rhodobacteraceae (*Tepidibacter mesophilus* and *Polaribacter huanghezhanensis*).

(ii)Aposymbiotic anemones versus anemones harbouring heterologous symbionts

A total of 38 differentially abundant bacterial taxa were recorded between aposymbiotic anemones and anemones symbiotic with *D. trenchii*. Of these, 11 were more abundant in anemones symbiotic with *D. trenchii*, including *Erythrobacter gaetbuli*, 2 taxa of the genus *Limimaricola*, and 1 taxon from each of the orders Cytophagales and Rhizobiales. A total of 27 ASVs were more abundant in aposymbiotic anemones, including 8 taxa of the family Vibrionaceae, 3 taxa of the phylum Bacteroides, and 1 taxon of the family Campylobacteraceae.

(iii)Anemones harbouring homologous symbionts versus heterologous symbionts

A total of 14 differentially abundant bacterial taxa were recorded between anemones symbiotic with *B. minutum* versus *D. trenchii*. Of these, three taxa were more abundant in anemones symbiotic with *D. trenchii*, including two taxa of the family Rhodobacteriaceae, and one taxon of the family Vibrionaceae. In contrast, 11 were more abundant in anemones symbiotic with *B. minutum*, including 2 taxa of the family Alteromonadaceae, 2 taxa of the genus *Chlamydiia*, and 1 taxon of the genus *Labrenzia*.

Non-metric multidimensional scaling further demonstrated a clear distinction between the microbial communities associated with each of the symbiotic states (Figure 2). PERMANOVA (F_2,12_ = 3.704, *p* < 0.01) post hoc analysis indicated that all three symbiotic states were distinct from each other: aposymbiotic anemones harboured microbiota distinct from hosts symbiotic with *D. trenchii* (*p* < 0.05) and those symbiotic with *B. minutum* (*p* < 0.05), while anemones symbiotic with *B. minutum* were distinct from those symbiotic with *D. trenchii* (*p* < 0.05).

### 3.3. BVOC Emissions Are Affected by Symbiont Identity

The volatilome was significantly influenced by symbiotic state (PERMANOVA: F_2,21_ = 1.9501, *p* < 0.01; Figure 3A). Specifically, the volatilome of aposymbiotic anemones and anemones symbiotic with *B. minutum* was significantly distinct (post hoc *p* < 0.01), while the volatilome of anemones symbiotic with *D. trenchii* was not significantly different from that of either aposymbiotic anemones (*p* = 1) or anemones symbiotic with *B. minutum* (*p* = 0.123), i.e., it overlapped with both.

A total of 142 BVOCs were detected across the three symbiotic states (Figure 3B). Of these, 29 were produced solely by aposymbiotic anemones, 13 were produced solely by anemones in symbiosis with *B. minutum*, and 13 were produced solely by anemones in symbiosis with *D. trenchii*. Forty-five BVOCs were present in all samples and were therefore designated as ‘core compounds’. Of these core compounds, 35 were identified, with the most abundant chemical class being aromatic compounds, followed by BVOCs with multiple different functional groups (classified as diverse functional groups, DFGs). The only halogenated hydrocarbon core compound identified was dibromomethane. Interestingly, the volatilome of aposymbiotic anemones was the most diverse, with 107 BVOCs, whereas the volatilome of anemones symbiotic with *B. minutum* was the least diverse, with only 70 BVOCs. Anemones symbiotic with *D. trenchii* produced a volatilome comprised of 97 BVOCs. The most abundant chemical class was DFGs for aposymbiotic anemones (22) and *B. minutum*-populated anemones (17), and unclassified compounds (24) for *D. trenchii*-populated anemones (Figure 3C).

### 3.4. Contrasting Volatilomes between Symbiotic States

A total of eight BVOCs were differentially abundant between the three symbiotic states, as determined via pairwise differential abundance testing (Appendix A, Figure 4). Only six differentially abundant BVOCs were recorded between aposymbiotic anemones and Aiptasia symbiotic with *B. minutum*. Of these, isoprene and dimethyl sulphide were detected in higher quantities in the volatilome of *B. minutum*-populated anemones (*p* < 0.001 and <0.05, respectively), while octanal, dodecanal, nonanal, and cis-6-nonenol were detected in higher quantities in the volatilome of aposymbiotic anemones (*p* < 0.05 for all comparisons). A further three differentially abundant BVOCs were identified between aposymbiotic anemones and anemones containing *D. trenchii*. Dimethyl sulphide and 2-methoxy-thiazole were detected in higher quantities in the volatilome of anemones symbiotic with *D. trenchii* (*p* < 0.001 and <0.05, respectively), while 1,1,2,2,3,3-hexamethylindane was detected in higher quantities in the volatilome of aposymbiotic anemones (*p* < 0.01). Only one differentially abundant BVOC was detected between anemones symbiotic with *B. minutum* vs. *D. trenchii*; specifically, isoprene was detected in higher quantities in the presence of *B. minutum* (*p* < 0.05).

### 3.5. Correlation of Microbiome and Volatilome

The relationship between the microbiome and volatilome was explored using Procrustes correlation analysis (Appendix A) to assess the extent to which shifts in the microbial community rather than Symbiodiniaceae species might explain the observed altered BVOC patterns. Microbial diversity did not, however, significantly influence the volatilome in the three symbiotic states (*p* = 0.138).

## 4. Discussion

Cnidarian metabolism changes in response to symbiosis with different symbiont species [18], and BVOC analysis can provide a non-invasive technique with which to explore the underlying metabolic regulation and restructuring associated with this change (see [42]). We examined the microbial composition and BVOC output of the model cnidarian Aiptasia in three different symbiotic states: (1) aposymbiosis; (2) symbiosis with the homologous symbiont, *B. minutum*; and (3) symbiosis with a heterologous symbiont, *D. trenchii*. While it is important to acknowledge that the need to hold aposymbiotic anemones in prolonged darkness prior to analysis may have influenced their microbiome and volatilome relative to that of the two symbiotic states that were maintained on a diel light/dark cycle, our BVOC analysis was consistent with earlier gene expression [18], proteomic [21], and metabolomic [18] studies. Specifically, *D. trenchii* appeared to form a physiologically sub-optimal symbiosis with Aiptasia, with the response of *D. trenchii*-populated anemones sharing features with both aposymbiotic and *B. minutum*-populated anemones, while these two latter states were markedly different to one another. Here, we discuss the shift in the volatilome and microbiome in response to symbiosis with different species of Symbiodiniaceae, and the potential sources and functions of released BVOCs.

### 4.1. Symbiosis and Symbiont Type Are Correlated to Changes in the Holobiont Microbiome

Microbial communities shift in response to environmental change in a variety of systems, including humans [76], plants [77], corals [78], and anemones [79]. Indeed, the Aiptasia microbiome, which has been defined in multiple studies [35,80], has previously been shown to respond to both symbiotic state [31] and thermal stress [57]. Here, we demonstrate that the microbiome also shifts in response to symbiosis with a non-native, physiologically sub-optimal species of Symbiodiniaceae.

Across the three symbiotic states, a core microbiome was revealed that was comprised of ten bacterial taxa, including *Maricaulis maris* and the nitrogen fixer *Rhizobium subbaraonis*. The ubiquity of these taxa may highlight their importance to the cnidarian host, regardless of symbiotic state. An analysis of human-associated microbes (The Human Microbiome Project) revealed few common bacterial species across all of the individuals studied, and it was proposed that the ‘core’ microbiome be interpreted in terms of the functionality of metabolic pathways rather than the specific species present [81]. Although we observed a difference at the species level between our study and another study investigating the microbiome of symbiotic and aposymbiotic anemones [33], the respective microbiomes may not be functionally different from each other. Thus, these observed microbiome differences may have little consequence to holobiont function in the model systems between the experimental system of Röthig et al. [33] and ours. Although we observed a difference at the species level, the only bacterial taxon common to the two studies was *Alteromonas* sp., with this dissimilarity perhaps being unsurprising given that both studies used laboratory Aiptasia cultures reared over many generations in an isolated environment. The repeated presence of *Alteromonas* sp. in the microbiome of Aiptasia is consistent with this bacterium playing an important role in the holobiont community and warrants further exploration.

Our data also show that anemones symbiotic with the homologous symbiont are associated with a more diverse microbiota relative to aposymbiotic anemones and those containing heterologous symbionts. This diversity could be indicative of a healthier, more balanced holobiont. Indeed, in humans, a diverse gut microbiome correlates with improved personal health, with links to a more nutritious diet [82,83]. A link between the microbiome and holobiont health is also suggested by the greater abundance of bacteria belonging to the family Vibrionaceae in aposymbiotic anemones, relative to anemones containing either Symbiodiniaceae species. *Vibrio* species have been identified as being prominent coral pathogens and implicated in coral disease. For example, *Vibrio coralliilyticus* is a known pathogen of the coral *Pocillopora damicornis*, in which infection causes cell lysis in corals at high temperatures [84,85]. Similarly, *Vibrio shiloi* is thought to cause bleaching in the coral *Oculina patagonica* by secreting extracellular materials that inhibit photosynthesis [86], while *Vibrio owensii* is particularly abundant in corals suffering from *Acropora* white syndrome [87]. Additionally, bacteria from the family Campylobacteraceae, which were detected in high levels in aposymbiotic anemones, are known pathogens in humans [88]. However, it is unknown as to whether bacteria from this family can be pathogenic to cnidarians. The reasons for a microbiome shift in aposymbiotic anemones towards reduced diversity and putatively more pathogenic bacteria are unknown, but could be related to a poor nutritional state due to the absence of photosynthate provided by the dinoflagellate symbionts and compromised defences against pathogens. For example, the surface microbes of corals are regulated, in part, through the periodic sloughing of mucus [89], yet much of this mucus is synthesised from the products of symbiont photosynthesis [90]; so, it seems reasonable to hypothesise that aposymbiotic anemones will produce less mucus than symbiotic ones which could impact the microbial diversity. Furthermore, there is both transcriptomic and proteomic evidence to suggest that aposymbiotic anemones exhibit higher levels of cellular oxidative stress than symbiotic anemones [18,91]. Likewise, the observed decrease in microbial species richness between anemones with homologous symbionts versus those with heterologous symbionts may indicate the presence of a sub-optimal dinoflagellate symbiont. Crocinitomicaceae, previously classified within the Cryomorphaceae family [92], were particularly common when anemones harboured *D. trenchii*; these bacteria are often found in areas rich in organic carbon, including in seawater, marine sediment, and coral mucus [92]. Why they were more abundant in the presence of the heterologous dinoflagellate is unknown, but it is plausible that they simply benefitted from reduced competition from other members of the microbiome.

### 4.2. Symbiosis and Symbiont Types Induce Changes in the Holobiont Volatilome

BVOC emissions are being increasingly explored as indicators of the health of organisms, including humans [93], and the health of ecosystems [94], including coral reefs [42,95]. Additionally, BVOCs are recognised as potential signalling molecules between phylogenetically distinct organisms [3,96]. Indeed, as has been proposed for other organically produced molecules like primary metabolites [18] and proteins [21], BVOCs may play a role in cellular signalling in the cnidarian-dinoflagellate symbiosis. For example, BVOCs could act as messengers to other individuals, perhaps to signal stress to neighbouring cnidarians, to induce an immune response, or to attract a particular symbiont species. Conversely, neighbouring cnidarians could be ‘warned’ against a less desirable algal partner through the triggering of an immune response in neighbouring cnidarians, as has been observed with parasitic insects via BVOC production by terrestrial plants [97,98]. Crucially, in this regard, our analysis showed that microbial diversity and the volatilome were not significantly correlated, suggesting that the bulk of the BVOCs detected were of anemone or Symbiodiniaceae origin.

Dimethyl sulphide (DMS) was found in high abundance in the volatilome of both Aiptasia-Symbiodiniaceae combinations, irrespective of symbiont identity, but was absent in the volatilome of aposymbiotic anemones. This contrasts with our previous study in which we did identify small amounts of DMS in the volatilome of aposymbiotic anemones [43], potentially arising from the bacterial metabolism of dimethylsulphoniopropionate (DMSP) to DMS and acrylate [99]; nevertheless, in all reported cases, the presence of Symbiodiniaceae is associated with a much more prolific release of DMS from the holobiont, highlighting a central role for the algal partner in its synthesis [43,100]. Moreover, we found DMS production to be greatest in the presence of the heterologous *D. trenchii*. One explanation for this could relate to innate physiological differences between the two symbiont types, as previous work on DMS production showed that *D. trenchii* is a more prolific producer of DMS than *Breviolum* sp. [39]. A second explanation, however, which is not mutually exclusive, could relate to the finding that *D. trenchii* elicits more oxidative stress in Aiptasia than *B. minutum* [18]. DMSP, the precursor of DMS, has multiple roles in other ecosystems, including osmoregulation [101], cryoprotection [102], as a foraging cue for fishes, and as an attractant for a diversity of marine bacteria (e.g., [103]); but, of particular interest here, it is known as an effective scavenger of reactive oxygen species (ROS). Furthermore, its breakdown products, DMS and acrylate, are 20–60 times more reactive than DMSP [103], and so are even more effective at scavenging ROS. Together, these three compounds (DMSP, DMS, and acrylate) act as a powerful antioxidant system, and could help to combat cellular stresses induced by the non-native symbiont.

Isoprene was detected in higher abundance in anemones symbiotic with *B. minutum* than in aposymbiotic anemones or anemones populated with *D. trenchii*. This prolifically emitted BVOC has functions in thermal tolerance in the tropical tree *Vismia guianensis* [104], scavenging ROS in the tall grass *Phragmites australis* [105], and providing protection against herbivory in the tobacco plant *Nicotiana tabacum* by deterring potential plant pests [106]. Although isoprene has been detected from microalgae—including diverse members of the Symbiodiniaceae—previously [107], to our knowledge, this is the first report on its production in Aiptasia. Should the protective function be conferred to the cnidarian host, these results may indicate that symbiosis with *B. minutum* conveys higher thermal tolerance in symbiosis with this native symbiont relative to symbiosis with the non-native *D. trenchii* or aposymbiosis, although these protective effects could be due to higher symbiont cell density in anemones symbiotic with *B. minutum* relative to those symbiotic with *D. trenchii*, rather than an effect due to symbiont identity alone.

Another compound, nonanal, was detected at significantly higher levels in the volatilome of both aposymbiotic anemones and anemones containing heterologous symbionts relative to those with homologous symbionts. It is possible that nonanal could act as an infochemical in cnidarians signalling a non-optimal symbiotic state. Nonanal is produced by a variety of plant species and has been shown to elicit a response in both insects and neighbouring plants. For example, nonanal production by the cactus *Opuntia stricta* acts as an attractant to the South American cactus moth *Cactoblastus cactorum* [108]. Furthermore, the emission of nonanal by barley (*Hordeum vulare*) and lima beans (*Phaseolus lunatus*) infected with the fungus *Blumeria graminis* and bacterium *Pseudomonas syringae,* respectively, induced systemic-acquired resistance in neighbouring plants [96,109], priming the immune system of uninfected plants by acting as a chemical warning signal. Nonanal is also emitted during courtship by the female vector for chagas disease *Triatoma infestans* to attract male suitors [110], and acts as an aggression pheromone in the bed bug *Cimex lectularius* [111]. There is therefore considerable potential for nonanal to play a comparable role as an infochemical in the Aiptasia system, making it an interesting candidate for future research. Though the signalling pattern of nonanal is common with 6-cis-nonenal, dodecanal, and octanal, their synthesis pathways are unknown. Their shared pattern of expression, however, could suggest that they share a common signalling pathway. Given that they were detected in both sub-optimal symbiotic states (aposymbiosis and symbiosis with *D. trenchii*), these molecules could be upregulated during periods of stress, and shut down in the optimal symbiotic state (i.e., symbiosis with *B. minutum*).

Other chemicals detected in higher abundance in aposymbiotic anemones relative to those in symbiosis with *B. minutum* include dodecanal and octanal. Dodecanal has been identified as a sex pheromone in ring-tailed lemurs [112], while in the moth *Zygaena filipendulae* it is a by-product of metabolism or stress [113]. Its release by aposymbiotic anemones could therefore be consistent with either a role in the chemoattraction of symbionts or host stress metabolism. In comparison, octanal is released from insect-damaged potato tubers and has putative roles in the attraction of nematode predators [114]. If this compound similarly plays a role in pathogen defence in Aiptasia, then this pattern of volatile production could reflect the putative shift towards a more pathogenic microbiome in aposymbiotic anemones, as discussed above, which has also been observed in bleached corals [115].

Collectively, we found that the volatilome of Aiptasia had a reduced diversity of compounds in the presence of the homologous *B. minutum*. This perhaps indicates a tighter recycling of metabolic products between the cnidarian host, dinoflagellate symbiont, and the associated microbiome, as may be expected for an optimally functional holobiont. In contrast, the greater diversity of the volatilome of anemones containing the heterologous *D. trenchii* and especially aposymbiotic anemones perhaps arises from a lesser degree or lack of inter-partner metabolic integration, respectively, and even cellular stress. Previous work on nutritional interactions in the Aiptasia-dinoflagellate symbiosis has shown that heterologous symbionts may form a metabolically sub-optimal, less-well-integrated association when compared to homologous symbionts [18,55,116,117,118]. Further work is needed to deduce the cause of variability observed between biological replicates, both within and between studies. Work is also needed to elucidate the BVOC pathways involved and how the various partners of the holobiont, including members of the microbiome, interact to metabolise and modify different volatiles to generate the BVOC emission patterns observed.

## 5. Conclusions

In conclusion, we have shown that symbiosis with the non-native dinoflagellate symbiont *D. trenchii* induces a distinct shift in both the microbiome and volatilome of the Aiptasia holobiont. The volatilomes of aposymbiotic anemones and those populated with the homologous *B. minutum* were distinct from one another, and yet both shared characteristics with the volatilome of *D. trenchii*-populated anemones, consistent with the view that *D. trenchii* forms a physiologically sub-optimal symbiosis with this host species. The ways in which different members of the holobiont contribute to the volatilome remain unclear given the complex inter-partner interactions involved. To elucidate these interactions, greater insight is needed into the functional roles for specific BVOCs to reconcile the organismal and ecological implications of any changes in intracellular symbiont populations. Our study provides a foundation for these future investigations, as we aim to understand the implications of climate change on the cnidarian-dinoflagellate symbiosis and coral reef function and health.

## Figures and Tables

**Figure 1 biology-12-01014-f001:**
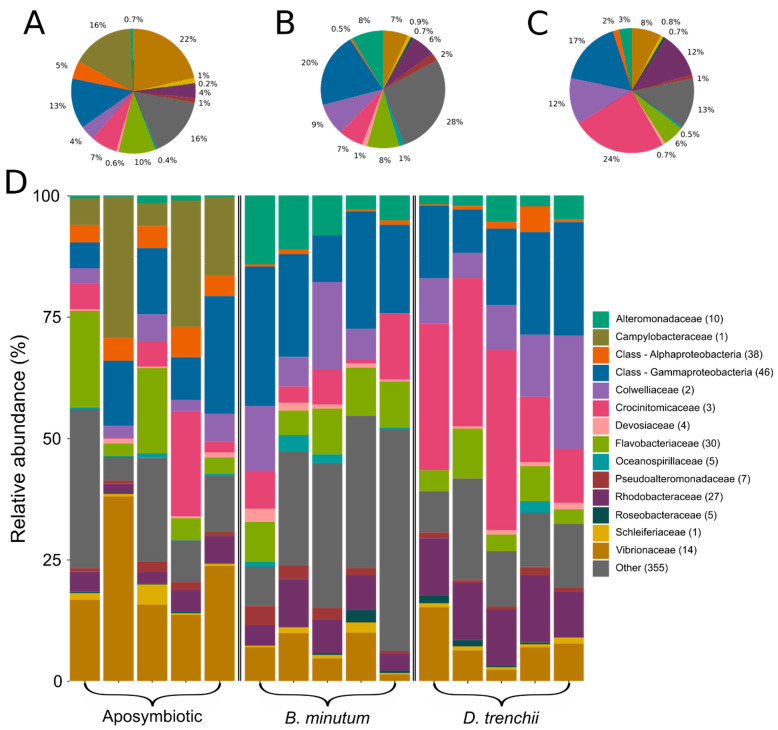
Bacterial community composition grouped by family. Each colour represents one of the most abundant 14 families identified across the dataset. Sequences unclassified at the family level were denoted at the class level. Less abundant families were designated as ‘other’. Numbers in parentheses denote the number of unique taxa within that family. Mean abundances across biological replicates are shown in (**A**) aposymbiotic anemones; (**B**) anemones symbiotic with homologous symbiont *Breviolum minutum*; and (**C**) anemones symbiotic with heterologous symbiont *Durusdinium trenchii*. (**D**) Relative abundances of bacterial taxa in each biological replicate.

**Figure 2 biology-12-01014-f002:**
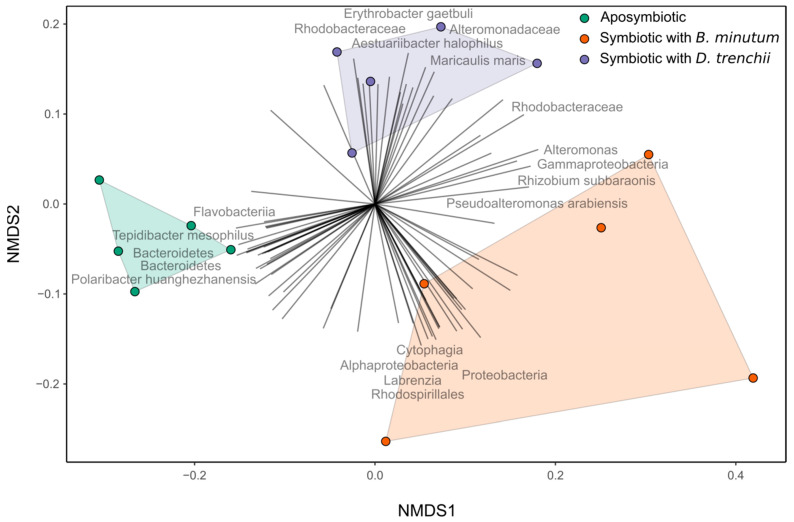
Non-metric multidimensional scaling (stress = 0.088) plot of bacterial amplicon sequence variants (ASVs) in three different symbiotic states in the Aiptasia model system. Displayed taxa were chosen based on the top 5 and bottom 5 loading scores for each NMDS dimension. PERMANOVA (*p* < 0.01); post hoc results indicated that all symbiotic states were distinct from each other.

**Figure 3 biology-12-01014-f003:**
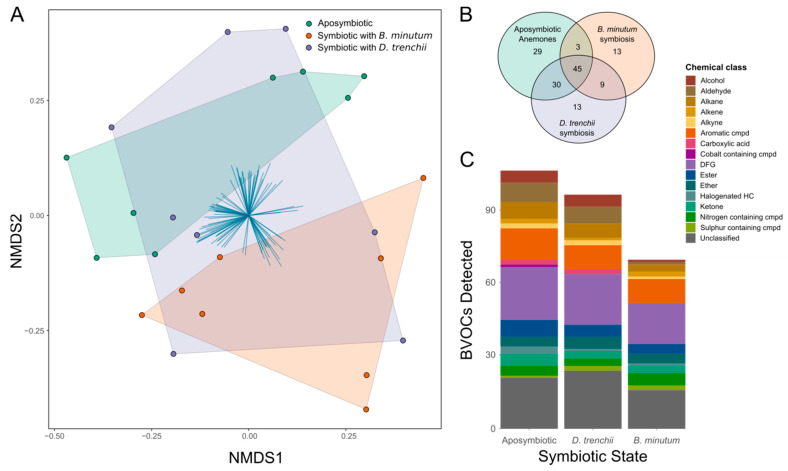
BVOC diversity among three symbiotic states in Aiptasia: aposymbiotic anemones, anemones symbiotic with the homologous symbiont (*Breviolum minutum*), and anemones symbiotic with a heterologous symbiont (*Durusdinium trenchii*). (**A**) Non-metric multidimensional scaling (NMDS) ordination plot of BVOCs detected across symbiotic states; (**B**) Venn diagram showing presence of 142 BVOCs detected across dataset; and (**C**) BVOCs grouped by chemical class. BVOCs had to be present in at least four of eight biological replicates within one symbiotic state to be included. DFG = diverse functional group; halogenated HC = halogenated hydrocarbon; cmpd = compound. See Appendix A for a complete list of the BVOCs detected.

**Figure 4 biology-12-01014-f004:**
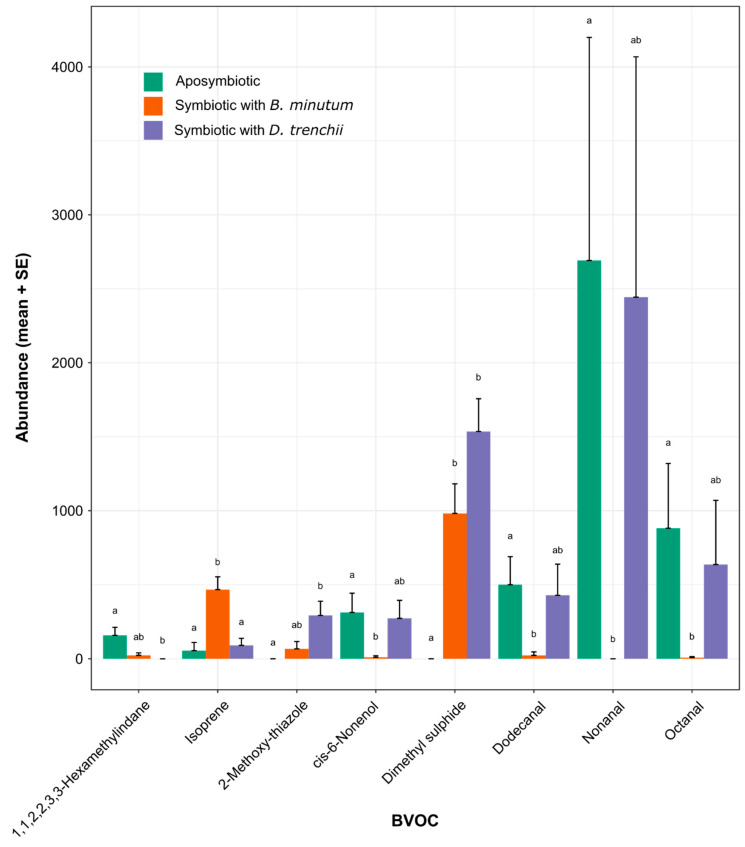
Differentially abundant BVOCs (+standard error) produced by aposymbiotic anemones, anemones symbiotic with *Breviolum minutum*, and anemones symbiotic with *Durusdinium trenchii*. Abundance data represent dimensionless areas under a peak normalised to chemical standard, seawater blanks, and protein content. Statistical significance (*p* < 0.05) is indicated by letters above error bars on the plot.

## Data Availability

Microbiome data are available at maggiewuerz/symbiont_identity_microbiome (github.com) (accessed on 23 June 2023).

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
