# Peer review of "Symbiont Identity Impacts the Microbiome and Volatilome of a Model Cnidarian-Dinoflagellate Symbiosis"

_biology, 2023, doi:10.3390/biology12071014_

Round 1

Reviewer 1 Report

This interesting study takes advantage of the actiniarian anemone “Aiptasia” to explore issues that may be relevant to other hexacorals such as reef-building corals.  In any event, the authors more-or-less assume that this is the case.  Building on a related paper (Wuerz et al. 2022, JEB), GC-MS was applied to elucidate the “volatilome” of Aiptasia.  Also, using rRNA sequences, various microbial symbionts were characterized.  While the results are interesting, there is an unfortunate methodological flaw.  As pointed out below, the aposymbiotic Aiptasia were grown in the dark, while the symbiotic ones were grown in the light.  How much of an influence this may have had on the data is not considered or at least not considered prominently enough.  It should be.  It would not be inappropriate to include the phrase “aposymbiotic anemones grown in the dark” throughout the manuscript.  Certainly, the effects of light should figure prominently in the discussion.  Some related and unrelated comments follow.

Abstract: “Aposymbiotic anemones produced a volatilome of 107 BVOCs, D. trenchii-populated anemones 97 BVOCs, and B. minutum-populated anemones 70 BVOCs. A total of six BVOCs were differentially abundant between aposymbiotic anemones and anemones symbiotic with B. minutum; three between aposymbiotic anemones and anemones symbiotic with D. trenchii; and one between anemones symbiotic with B. minutum and those containing D. trenchii.”  Were the aposymbiotic anemones cultured under the same conditions as the symbiotic ones?

Line 153-154: “aposymbiotic anemones were maintained in the dark at 25 153 °C for 1.5 years prior to BVOC sampling,” i.e., no

Line 178-180: “Symbiotic anemones were maintained at a constant temperature of 25 °C and light intensity of 70 µmol photons m-2 s -2 on a 12:12 h light:dark cycle as described above. Aposymbiotic anemones were maintained at a constant temperature of 25 °C and kept in the dark.”

Line 207: “amplification of the of the variable rRNA regions and sequencing of the amplification”: delete repeated words.

Line 232-235: “The night before BVOC sampling, experimental organisms were transferred into sterile 150 mL glass vials (Wheaton, Millville, NJ, USA) containing 75 mL 0.22 µm FSW, where they were retained under conditions identical to those used for growth,” i.e., still in the dark.

Line 320-326: “Aposymbiotic anemones were generally dominated by Vibrionaceae (average 22% of the microbiome) and Campylobacteraceae (16%), although variability among the samples was high. In contrast, anemones symbiotic with D. trenchii were dominated by Crocinitomicaceae (24%) and the class Gammaproteobacteria (mean 17%), and anemones symbiotic with B. minutum were dominated by the class Gammaproteobacteria (20%) and by ‘other’ bacteria (unclassified or rare taxa; 28%). Interestingly, a single ASV of Campylobacteraceae was identified exclusively in aposymbiotic anemones.”  Were some of the symbiotic bacteria (e.g., gammaproteobacteria) phototrophs?

Line 556-564: “A second explanation however, which is not mutually exclusive – could relate to the finding that D. trenchii elicits more oxidative stress in Aiptasia than B. minutum [18]. DMSP, the precursor of DMS, has multiple roles in other ecosystems, including osmoregulation [103], cryoprotection [104], and as a foraging cue for fishes and attractant for a diversity of marine bacteria (e.g., [105], but of particular interest here, it is known as an effective scavenger of reactive oxygen species (ROS). Fur- thermore, its breakdown products, DMS and acrylate, are 20–60 times more reactive than DMSP [106] and so even more effective at scavenging ROS. Together, these three compounds (DMSP, DMS, acrylate) act as a powerful antioxidant system, and could help combat cellular stresses induced by the non-native symbiont.”  Is it that they are non-native or are Durusdinium just high producers of ROS even if they are native?  I think the latter possibility has been suggested elsewhere in the literature.

Author Response

This interesting study takes advantage of the actiniarian anemone “Aiptasia” to explore issues that may be relevant to other hexacorals such as reef-building corals.  In any event, the authors more-or-less assume that this is the case.  Building on a related paper (Wuerz et al. 2022, JEB), GC-MS was applied to elucidate the “volatilome” of Aiptasia.  Also, using rRNA sequences, various microbial symbionts were characterized.  While the results are interesting, there is an unfortunate methodological flaw.  As pointed out below, the aposymbiotic Aiptasia were grown in the dark, while the symbiotic ones were grown in the light.  How much of an influence this may have had on the data is not considered or at least not considered prominently enough.  It should be.  It would not be inappropriate to include the phrase “aposymbiotic anemones grown in the dark” throughout the manuscript.  Certainly, the effects of light should figure prominently in the discussion.  Some related and unrelated comments follow.

Abstract: “Aposymbiotic anemones produced a volatilome of 107 BVOCs, D. trenchii-populated anemones 97 BVOCs, and B. minutum-populated anemones 70 BVOCs. A total of six BVOCs were differentially abundant between aposymbiotic anemones and anemones symbiotic with B. minutum; three between aposymbiotic anemones and anemones symbiotic with D. trenchii; and one between anemones symbiotic with B. minutum and those containing D. trenchii.”  Were the aposymbiotic anemones cultured under the same conditions as the symbiotic ones?

Reply: We maintained anemones in the dark in attempt to ensure their maintenance as aposymbiotic through the experiment, as per the standard approach for this model system, though we of course recognise that this causes some (unavoidable) inconsistency in the experimental approach. To address this directly in the manuscript, we have added the following statement: “While it is important to acknowledge that the need to hold aposymbiotic anemones in prolonged darkness prior to analysis may have influenced their microbiome and volatilome relative to that of the two symbiotic states that were maintained on a diel light/dark cycle, our BVOC analysis was consistent with earlier gene expression [18], proteomic [21] and metabolomic [18] studies.” (Lines 456 – 460).

Line 153-154: “aposymbiotic anemones were maintained in the dark at 25 153 °C for 1.5 years prior to BVOC sampling,” i.e., no

Reply: That’s correct – we consider this a necessary limitation to our experimental design, and we cannot rule out differences in metabolism or bacterial composition due to differences in light availability, as now addressed as highlighted above.

Line 178-180: “Symbiotic anemones were maintained at a constant temperature of 25 °C and light intensity of 70 µmol photons m-2 s -2 on a 12:12 h light:dark cycle as described above. Aposymbiotic anemones were maintained at a constant temperature of 25 °C and kept in the dark.”

Line 207: “amplification of the of the variable rRNA regions and sequencing of the amplification”: delete repeated words.

Reply: Done.

Line 232-235: “The night before BVOC sampling, experimental organisms were transferred into sterile 150 mL glass vials (Wheaton, Millville, NJ, USA) containing 75 mL 0.22 µm FSW, where they were retained under conditions identical to those used for growth,” i.e., still in the dark.

Reply: That’s correct – aposymbiotic anemones were held continuously in the dark to prevent the reestablishment of symbiosis as confirmed, and now clarified, as described above.

Line 320-326: “Aposymbiotic anemones were generally dominated by Vibrionaceae (average 22% of the microbiome) and Campylobacteraceae (16%), although variability among the samples was high. In contrast, anemones symbiotic with D. trenchii were dominated by Crocinitomicaceae (24%) and the class Gammaproteobacteria (mean 17%), and anemones symbiotic with B. minutum were dominated by the class Gammaproteobacteria (20%) and by ‘other’ bacteria (unclassified or rare taxa; 28%). Interestingly, a single ASV of Campylobacteraceae was identified exclusively in aposymbiotic anemones.”  Were some of the symbiotic bacteria (e.g., gammaproteobacteria) phototrophs?

Reply: It is indeed possible that some of the Gammaproteobacteria detected in symbiotic anemones are aerobic anoxygenic phototrophic bacteria, but we cannot determine this from the resolution of our data. While it is possible that there could be differences in the microbiome of aposymbiotic vs. symbiotic anemones that are not due to symbiosis, and instead due to the preparation of these organisms, the fact that the microbiome of anemones symbiotic with B. minutum vs. those symbiotic with D. trenchii were distinct suggests that pre-treatment differences between anemones were not sufficient to describe the observed differences in their microbiota.

Line 556-564: “A second explanation however, which is not mutually exclusive – could relate to the finding that D. trenchii elicits more oxidative stress in Aiptasia than B. minutum [18]. DMSP, the precursor of DMS, has multiple roles in other ecosystems, including osmoregulation [103], cryoprotection [104], and as a foraging cue for fishes and attractant for a diversity of marine bacteria (e.g., [105], but of particular interest here, it is known as an effective scavenger of reactive oxygen species (ROS). Fur- thermore, its breakdown products, DMS and acrylate, are 20–60 times more reactive than DMSP [106] and so even more effective at scavenging ROS. Together, these three compounds (DMSP, DMS, acrylate) act as a powerful antioxidant system, and could help combat cellular stresses induced by the non-native symbiont.”  Is it that they are non-native or are Durusdinium just high producers of ROS even if they are native?  I think the latter possibility has been suggested elsewhere in the literature.

Reply: Yes, this is true! D. trenchii has been shown to be a more prolific producer of DMS than Breviolum sp. We have now added a sentence to the discussion (with a relevant citation) to reflect this fact: “One explanation for this could relate to innate physiological differences between the two symbiont types, as previous work on DMS production showed that D. trenchii is a more prolific producer of DMS than Breviolum sp. [39].” (Lines 545 – 547).

Reviewer 2 Report

The manuscript by Wuerz et al. investigated the impacts of the symbiotic state (aposymbiotic vs. symbiotic) and the symbiont identity on the microbiome and biogenic volatile organic compounds production in the model Aiptasia system. The manuscript is globally well written, the experiments and design appear to be done well, and this is a highly effective group of researchers who are well-versed in this kind of work. Nevertheless, there are two points I would like to raise about this manuscript.

First, I think the authors need to discuss deeper into the issue of variability/repeatability related to BVOCs measurements. Indeed, the variability between biological replicates is very large and this is discussed nowhere in the present manuscript. Likewise, looking in more detail at the data from a recently published study by the same group in which they use a similar experimental system (analysis of BVOCs on Aiptasia symbiotic with B. minutum and aposymbiotic individuals, Wuerz et al. 2022. Journal of Experimental Biology, 225(19), jeb244600), I have found that the overlap between data in the published paper and those reported in the present manuscript was low. For instance, in the present manuscript, isoprene was detected in higher quantities in the volatilome of 1) the B. minutum-populated anemones compared to aposymbiotic anemones and 2) anemones symbiotic with B. minutum vs. D. trenchii, and this has been discussed in the context of higher thermal tolerance. But, in the published study, isoprene was not detected B. minutum-populated anemones ! For me, this raises a big repeatability issue related to BVOCs measurements, that needs to be discussed in this manuscript and addressed in the future if we are to progress in this field of research.

My second criticism is related to the content of the manuscript. As it stands, if the datasets related to the microbiome or BVOCs had been published separately, the result would have been substantially the same because in the present manuscript, the connections established by the authors between the two datasets are too few and the impact of the microbiome on BVOCs is clearly not discussed enough nor highlighted.

Here are my minor comments:

L224: information about the sequencing is missing.

L293-305: Are the sequences available in a public database? If yes, provide the references.

L299: add a space between “2022” and “from”.

L458-460: "Our analysis of BVOC emission profiles was consistent with these earlier observations, again showing that the heterologous D. trenchii induces an intermediate physiological state between that of B. minutum-populated and aposymbiotic anemones."

On what results obtained at the level of the BVOCs do you base yourself to affirm that the Aiptasia in symbiosis with D. trenchii are in an intermediate physiological state between aposymbiotic individuals and individuals living in symbiosis with B. minutum?

L511-513: Since representatives of the Family Campylobacteraceae are often commensals of warm-blooded animals, have you considered the possibility that the presence of these organisms in aposymbiotic anemones could result from external contamination at a given moment during your study (anemones raising, DNA extraction...)?

L624-625: “In conclusion, we have shown that symbiosis with the non-native dinoflagellate symbiont D. trenchii induces a distinct shift in the microbiome that may contribute to the observed shift in the volatilome” Or may not… You don't report or discuss results that would underpin this conclusion.

Author Response

The manuscript by Wuerz et al. investigated the impacts of the symbiotic state (aposymbiotic vs. symbiotic) and the symbiont identity on the microbiome and biogenic volatile organic compounds production in the model Aiptasia system. The manuscript is globally well written, the experiments and design appear to be done well, and this is a highly effective group of researchers who are well-versed in this kind of work. Nevertheless, there are two points I would like to raise about this manuscript.

First, I think the authors need to discuss deeper into the issue of variability/repeatability related to BVOCs measurements. Indeed, the variability between biological replicates is very large and this is discussed nowhere in the present manuscript. Likewise, looking in more detail at the data from a recently published study by the same group in which they use a similar experimental system (analysis of BVOCs on Aiptasia symbiotic with B. minutum and aposymbiotic individuals, Wuerz et al. 2022. Journal of Experimental Biology, 225(19), jeb244600), I have found that the overlap between data in the published paper and those reported in the present manuscript was low. For instance, in the present manuscript, isoprene was detected in higher quantities in the volatilome of 1) the B. minutum-populated anemones compared to aposymbiotic anemones and 2) anemones symbiotic with B. minutum vs. D. trenchii, and this has been discussed in the context of higher thermal tolerance. But, in the published study, isoprene was not detected B. minutum-populated anemones ! For me, this raises a big repeatability issue related to BVOCs measurements, that needs to be discussed in this manuscript and addressed in the future if we are to progress in this field of research.

Reply: We appreciate this comment, however metabolism studies generally are highly variable, and given that BVOCs are downstream products of primary metabolism, variability in volatilomics is to be expected. In anticipation of this, we used the highest number of biological replication possible given logistical constraints (e.g. mass spectrometer availability): n = 8 for each symbiotic state. By comparison, previous volatilomics studies (e.g., our previous work: Wuerz et al., 2022) used only n = 5. Further, it is important to note that the methods of detection for BVOCs were different between these two studies due to covid travel restrictions being implemented in between them. Our previous work separated mixtures of BVOCs using GCxGC-MS, while the dataset in the current manuscript was analysed using GC-MS. The GCxGC-MS provides a higher degree of sensitivity due to sample separation on two consecutive columns, while the GC-MS separates BVOCs using only one column. This means that great caution should be exercised when comparing across these studies. Lastly, we do not know the age of any anemones sampled, and such features could also cause of variability between studies. Nevertheless, the marked differences between the different treatments seen here, despite the inter-sample variability, highlight the robustness of our conclusions.

Rather than speculate extensively on the possible, unknown reasons for this variability in the manuscript, we have added a simple statement that: “Further work is needed to deduce the cause of variability observed between biological replicates, both within and between studies”. (Lines 608 – 609).

My second criticism is related to the content of the manuscript. As it stands, if the datasets related to the microbiome or BVOCs had been published separately, the result would have been substantially the same because in the present manuscript, the connections established by the authors between the two datasets are too few and the impact of the microbiome on BVOCs is clearly not discussed enough nor highlighted.

Reply: Thank you for the comment – we completely agree and have consulted with microbiologist colleagues about how best to link the two datasets. As a result, we have now added an additional analysis enabling the correlation of the two datasets. Specifically, Procrustes analysis was used, which is a analytical method that compares the distance between two objects by matching the distances between paired samples based on their NMDS scores. This analysis revealed that the microbiome and volatilome datasets were not significantly correlated, and thus that the microbiota associated with these symbiotic states were unlikely to have generated the different BVOC diversity patterns observed. Rather, the bulk of the BVOCs produced were more likely of anemone or Symbiodiniaceae origin.

We have now included a description of this correlation, and the resulting data in the manuscript, with a new subsection in the results entitled: “Correlation of microbiome and volatilome”. This section states: “The relationship between the microbiome and volatilome was explored using Procrustes correlation analysis (Fig. S2), to assess the extent to which shifts in the microbial community rather than Symbiodiniaceae species might explain the altered BVOC patterns observed. Microbial diversity did not, however, significantly influence the volatilome in the three symbiotic states (p = 0.138).” (Lines 445 – 448). We have also included a plot of this analysis in the supplementary information.

Furthermore, in the discussion we now state: “Crucially in this regard, our analysis showed that microbial diversity and the volatilome were not significantly correlated, suggesting that the bulk of the BVOCs detected were of anemone or Symbiodiniaceae origin.” (Lines 533 – 535).

Here are my minor comments:

L224: information about the sequencing is missing.

Reply: We have included more detail on the sample preparation, and have included the Illumina reference protocol that was used. This section now reads: “Samples were gently mixed and incubated at room temperature for 5 minutes, after which the supernatant was discarded. The same beads were resuspended with 80% ethanol, twice. After both ethanol cleanup steps, the supernatant was discarded and the beads allowed to air dry for 10 min. Once dry, beads were resuspended with 52.5 µL of 10 mM Tris pH 8.5 and incubated at room temperature for two minutes. Samples were then loaded onto the Illumina MiSeq platform for sequencing, according to the manufacturer’s protocol [59].” (Lines 215 – 220).

L293-305: Are the sequences available in a public database? If yes, provide the references.

Reply: Sequencing data are now publicly available on Github at the following link: https://github.com/maggiewuerz/symbiont_identity_microbiome.git. We have now referred to this link in the results section of our manuscript: “Sequencing data, along with taxonomic assignment, are publicly available at: http://github.com/maggiewuerz/symbiont_identity_microbiome.git.” (Lines 298 – 300).

L299: add a space between “2022” and “from”.

Reply: Done.  

L458-460: "Our analysis of BVOC emission profiles was consistent with these earlier observations, again showing that the heterologous D. trenchii induces an intermediate physiological state between that of B. minutum-populated and aposymbiotic anemones."

On what results obtained at the level of the BVOCs do you base yourself to affirm that the Aiptasia in symbiosis with D. trenchii are in an intermediate physiological state between aposymbiotic individuals and individuals living in symbiosis with B. minutum?

Reply: Based on our PERMANOVA analysis of our multivariate data, the volatilomes of aposymbiotic anemones and anemones symbiotic with B. minutum were distinct from each other; anemones symbiotic with D. trenchii were statistically indistinct from either aposymbiotic anemones or anemones symbiotic with B. minutum. We have removed all references to symbiosis with a heterologous symbiont as producing an “intermediate” volatilome, as we agree that this terminology is potentially misleading.

L511-513: Since representatives of the Family Campylobacteraceae are often commensals of warm-blooded animals, have you considered the possibility that the presence of these organisms in aposymbiotic anemones could result from external contamination at a given moment during your study (anemones raising, DNA extraction...)?

Reply: We cannot rule this out, though it seems unlikely. The amount of anemone handling was the same across symbiotic states; aposymbiotic anemones received the same level of handling as both symbiotic treatments during the experiment. Because of the consistency with which different groups of animals were handled, we are confident that any differentially abundant BVOCs between groups reflected true physiological differences between holobionts.  

L624-625: “In conclusion, we have shown that symbiosis with the non-native dinoflagellate symbiont D. trenchii induces a distinct shift in the microbiome that may contribute to the observed shift in the volatilome” Or may not… You don't report or discuss results that would underpin this conclusion.

Reply: In light of the correlation analysis performed (see above), we have removed this comment. The sentence now reads: “In conclusion, we have shown that symbiosis with the non-native dinoflagellate symbiont D. trenchii induces a distinct shift in the microbiome and volatilome of the Aiptasia holobiont.” (Lines 610 – 612).

Reviewer 3 Report

In this study, Wuerz et al. analyzed the effect of Symbiont identity on the microbiome and volatilome of the Aiptasia holobiont.  In order to attain this object, three different coral models were constructed. Aposymbiotic Aiptasia, Aiptasia associated with native Symbiodiniaceae (Breviolum minutum) and Aiptasia associated with non-native Symbiodiniaceae (Durusdinium trenchii).  After continous cultivation of these differen symbiotic models (about six weeks), they analyzed the microbiome (bacteria) and the Volatilome (biogenic Volatile Organic Compounds) in these models. they found distinct microbioal communities and intermediated state of the valatilome when Aiptasia hosted a non-native Symbiodiniaceae.  Such a bactiral and metabolic shifts might reflect the different responses of coral populated with varous different Symbiodiniaceae clades under global climate changes. In all, this study is well designated and well written, and this article is a very interesting study of coral-Symbiodiniacese-bacteria holobiont. It is valuable to be published to this journal. 

Author Response

In this study, Wuerz et al. analyzed the effect of Symbiont identity on the microbiome and volatilome of the Aiptasia holobiont.  In order to attain this object, three different coral models were constructed. Aposymbiotic Aiptasia, Aiptasia associated with native Symbiodiniaceae (Breviolum minutum) and Aiptasia associated with non-native Symbiodiniaceae (Durusdinium trenchii).  After continous cultivation of these differen symbiotic models (about six weeks), they analyzed the microbiome (bacteria) and the Volatilome (biogenic Volatile Organic Compounds) in these models. they found distinct microbioal communities and intermediated state of the valatilome when Aiptasia hosted a non-native Symbiodiniaceae.  Such a bactiral and metabolic shifts might reflect the different responses of coral populated with varous different Symbiodiniaceae clades under global climate changes. In all, this study is well designated and well written, and this article is a very interesting study of coral-Symbiodiniacese-bacteria holobiont. It is valuable to be published to this journal. 

Reply: Fantastic, thank you for the positive feedback!

Round 2

Reviewer 2 Report

I am satisfied with the answers and the modifications made by the authors and I recommend this manuscript for publication.